# Biocompatible Optical Fibers Made of Regenerated Cellulose and Recombinant Cellulose-Binding Spider Silk

**DOI:** 10.3390/biomimetics8010037

**Published:** 2023-01-15

**Authors:** Martin Reimer, Kai Mayer, Daniel Van Opdenbosch, Thomas Scheibel, Cordt Zollfrank

**Affiliations:** 1Chair for Biogenic Polymers, Technical University of Munich Campus Straubing for Biotechnology and Sustainability, Schulgasse 16, 94315 Straubing, Germany; 2Chair for Biomaterials, Faculty of Engineering Science, University of Bayreuth, Prof. Rüdiger-Bormann-Straße 1, 95447 Bayreuth, Germany; 3Bayreuther Zentrum für Kolloide und Grenzflächen (BZKG), Universität Bayreuth, Universitätsstraße 30, 95440 Bayreuth, Germany; 4Bayreuther Materialzentrum (BayMat), Universität Bayreuth, Universitätsstraße 30, 95440 Bayreuth, Germany; 5Bayreuther Zentrum für Molekulare Biowissenschaften (BZMB), Universität Bayreuth, Universitätsstraße 30, 95440 Bayreuth, Germany; 6Bayrisches Polymerinstitut (BPI), Universität Bayreuth, Universitätsstraße 30, 95440 Bayreuth, Germany

**Keywords:** attenuation, transmission, light guiding, data rate, photodynamic therapy, optogenetic stimulation

## Abstract

The fabrication of green optical waveguides based on cellulose and spider silk might allow the processing of novel biocompatible materials. Regenerated cellulose fibers are used as the core and recombinantly produced spider silk proteins eADF4(C16) as the cladding material. A detected delamination between core and cladding could be circumvented by using a modified spider silk protein with a cellulose-binding domain-enduring permanent adhesion between the cellulose core and the spider silk cladding. The applied spider silk materials were characterized optically, and the theoretical maximum data rate was determined. The results show optical waveguide structures promising for medical applications, for example, in the future.

## 1. Introduction

Polymer optical fibers (POFs) are guiding fibers for light and data transmission. They are based on a core surrounded by one or multiple claddings of lower refractive indices [1]. These fibers are predominantly used in short-distance applications, such as in electronic devices and medical applications, for example, photodynamic therapy (PDT) or optogenetic stimulation [2,3]. However, due to their production from fossil resources and processing conditions from harsh solvents or at high temperatures, it is desirable that more environmentally friendly and biocompatible approaches must be implemented [4]. One such system is described here, where for the first time, regenerated cellulose (RC) and recombinant spider silk are processed into a light waveguiding structure with cellulose as core and recombinant spider silk as a cladding material. With regard to environmental factors, the advantages of these materials are apparent, as cellulose is derived from regrowing plant material, and microorganisms can produce recombinant spider silk proteins in large-scale bioreactors [5]. The excellent biocompatibility of cellulose [6,7,8], combined with the biocompatibility and shielding properties of recombinant spider silk [9,10,11] make these novel optical fibers a promising alternative for medical applications.

Both materials generally have great potential for use in optoelectronic devices due to their optical properties. Here, the interaction of materials with light can be specifically modified depending on their use. Cellulose has been processed into artificially ordered structures, such as photonic crystals, chiral nematic liquid crystals and Bragg stacks, as well as disordered structures, such as random lasers to manipulate light in a targeted manner. Furthermore, cellulose and its derivatives are also applied in organic light-emitting diodes, flexible touch screens, solar cells, as well as optical fibers [12]. For optical fibers, Dupuis et al. prepared a porous dual-core fiber structure by using commercially available cellulose butyrate tubes (refractive index 1.475) with different diameters as core and shell material, respectively. The space between these tubes was then filled with a polydisperse hydroxypropyl cellulose powder (refractive index of 1.337) to yield a lower-index inner cladding and led to the fabrication of optical fibers from two types of biodegradable cellulose [13]. Orelma et al. processed cellulose directly into an optical fiber for water sensor applications. There, cellulose was dissolved in the ionic liquid [EMIM]OAc and coagulated in water to form a regenerated fiber by dry-jet wet-spinning. The regenerated cellulose fiber was then coated with cellulose acetate. The obtained biopolymer optical fibers (bioPOF) reached an attenuation minimum of 5.9 dB cm^−1^ at a wavelength of 1130 nm [14]. In a previous publication, we investigated the potential of cellulose-based optical fibers for advanced engineering applications. This was accomplished by dissolving microcrystalline cellulose in the binary solvent system consisting of DMAc/LiCl and then coagulating it in ethanol to form a fiber. The fibers were optically characterized both without and with cellulose acetate, cellulose acetate propionate, and cellulose acetate butyrate as coatings. Here, the fibers exhibited an attenuation minimum of 0.56–0.82 dB cm^−1^ at about 860 nm, depending on the cladding structure. In addition, the ultimate transmission loss limit of the cellulose optical fibers was determined. This showed that cellulose-based optical fibers can replace fossil-based fibers in the future if the extrinsic losses are reduced [15].

For silk-based optical structures, Omenetto et al. have shown that silkworm silk from *Bombyx mori* can be processed into free-standing biological matrices that show the material toughness to withstand room-temperature use under environmental conditions while simultaneously exhibiting high optical quality (about 92% across the visible range) [16]. Furthermore, Parker et al. demonstrated a directly bioprinted waveguide made from *Bombyx mori* silk protein hydrogels by simply printing it on a borosilicate glass substrate (refractive index 1.52) which showed a lower refractive index than the silk protein fiber (refractive index 1.54) [17]. Finally, Applegate et al. have demonstrated a way to produce biocompatible optical waveguides by combining a film stripe with a hydrogel, both made from the silk of *Bombyx mori* [18]. By combining a hydrogel morphology, with the main contributor to the refractive index being water, with a dry film, they produced a functional film/gel tube that successfully guided light. All these studies combined a similar approach of only using one component and the surrounding media as low refractive “shell”, which rendered these fibers less useful and very fragile in most environments and applications. Gel-like or powder structures are also not resistive against damage from outside. These are crucial parameters if these fibers should be used in technological or medical applications. These publications already suggested that cellulose and silk proteins are promising candidates for fabricating novel biocompatible optical waveguides, however, the literature so far only focused on using *Bombyx mori* silk proteins.

The literature covering cellulose in combination with recombinant spider silk proteins is sparse. Spider silk has been blended with cellulose to yield stronger fibers than the ones made from spider silk alone [19]. Two methods showed a cellulose-binding tag recombinantly attached to a spider silk protein but focused on the interaction with nano cellulose crystals in sponges. The findings supported the theory that combining both materials yields in a material whose mechanical properties surpass the ones of the individual materials [20,21]. Additionally, recombinant spider silk proteins pose an even more promising approach in optical applications than *Bombyx mori* silk. It is produced recombinantly, meaning that desired modifications could be built in to enhance optical and mechanical properties [22,23,24]. In this research, this exact property was the key to producing feasible lightwave guides using a spider silk protein with a cellulose-binding domain (CBD) designed and produced to yield a stable core-cladding structure of the cellulose core fibers. This leads to the first successful combination of regenerated cellulose and recombinant spider silk for the construction of an optical fiber.

## 2. Materials and Methods

### 2.1. Materials

Microcrystalline cellulose (MCC, with a degree of polymerization (DP) of 221, M_w_ = 35,800 g mol^−1^), lithium chloride (LiCl) and 99% *N*,*N*-dimethylacetamide (DMAc) were purchased from Merck KGaA (Darmstadt, Germany), VWR (Leuven, Belgium and Alfa Aesar (Kandel, Germany), respectively. Ethanol (technical, 99.6%) (EtOH) and ammonium bicarbonate (NH₄HCO₃) was acquired from Carl Roth GmbH & Co. KG (Karlsruhe, Germany). 1,1,1,3,3,3-hexafluoro-2-propanol (HFIP) was purchased from Th. Geyer GmbH & Co. KG (Renningen, Germany). eADF4(C16) was obtained from AMSilk (Neuried, Germany). All other chemicals were used as received with no further purification.

### 2.2. Methods

#### 2.2.1. Dissolution of MCC

MCC was dissolved in DMAc/LiCl according to our previous publication [15]. In brief, 5 g of freeze-dried MCC (30.9 mmol) was suspended in 100 mL of DMAc and stirred for 72 h at room temperature (RT). The cellulose suspension was slowly heated to 90 °C and stirred at this temperature for 1 h. 9.81 g of dried LiCl (231.4 mmol) was slowly added to the batch. The batch was then cooled by 0.53 °C per minute to RT to obtain a colorless cellulose solution.

#### 2.2.2. Wet-Spinning of Regenerated Cellulose Fibers

Wet-spinning was performed according to our previous publication with exception of the flow rate of the pump and the drying [15]. A laboratory syringe pump (LA-30, Landgraf Laborsysteme HLL GmbH, Germany) was used to prepare the core filament. For this purpose, the cellulose solution was flowed through a 15 cm long silicone tube (3.0 mm diameter × 1.6 mm wall thickness) into an 99.6% EtOH coagulation bath with a constant flow rate of 1.5 mL min^−1^. The freshly regenerated fiber was wound up with an electric rotary motor. The produced filaments were kept in the EtOH coagulation bath for at least 1 h. The completely regenerated cellulose fibers were washed with distilled water five times for 3 h to remove the components of the used solvent system. Afterwards, the fibers were attached to a hook and dried by hanging at ambient conditions stretched by a 12 g weight.

#### 2.2.3. Processing of eADF4(C16)-CBD

To generate a cellulose-binding recombinant spider silk protein, CBD_cex_, the genetic sequence of the c-terminal cellulose-binding domain of the exoglucanase enzyme in *Cellulomonas fimi* [25], was c-terminally added to the eADF4(C16) sequence via a seamless cloning approach as previously reported [26]. The resulting fusion protein with a total size of 57.1 kDa could be produced by the host organism *Escherichia coli* and purified via ion metal affinity chromatography (IMAC). Cellulose binding and further processing of the fusion protein could be demonstrated shown (unpublished data).

#### 2.2.4. Production of Recombinant Spider Silk Films

For the optical characterization of the recombinant spider silk material, silk films were prepared. For this purpose, lyophilized eADF4(C16) and eADF4(C16)-CBD powder was dissolved overnight in HFIP (8.5 mg mL^−1^). The solution was then centrifuged for 30 min at 12,000 rpm and cast in a cavity of an 8-well removable slide plate and dried overnight. For the post-treatment, the films were immersed in 70, 75 or 80 wt% EtOH for one minute and then air dried with corners fixed by adhesive tape. The transparent untreated and post-treated films could then be used for optical characterization.

#### 2.2.5. Coating with eADF4(C16)

For the fabrication of a core-cladding structured optical fiber, a freshly prepared eADF4(C16) solution in HFIP was dropwise applied to a regenerated cellulose filament. The droplet ran down the fiber and covered the filament. The solvent HFIP evaporated at RT. For the post-treatment, the cladded fibers were then immersed in a 75 wt% EtOH solution for one minute and afterwards dried in a desiccator for three days.

#### 2.2.6. Coating with eADF4(C16)-CBD

In order to ensure an active binding domain, the coating process had to be carried out in an aqueous solution. To enable uniform coating and to keep the cellulose fibers’ swelling at a minimum, custom-made fiber holders were used where the individual fibers were glued on at their ends. Cellulose fibers 15 cm long were coated with a 1 mg mL^−1^ solution of eADF4(C16)-CBD in 10 mM NH_4_HCO_3_ by horizontally introducing them to the protein solution for 2 min, followed by a 30 min drying period in air. For post-treated fibers, the dry and mounted fibers were then vertically stored in a desiccator in an ethanol atmosphere for further 30 min. To release the fibers from the holder, razor blades were used to cut behind the glued area.

#### 2.2.7. Ultraviolet and Visible Light Spectroscopy

For all optical characterizations and attenuation determinations, we used an ultraviolet and visible (UV/VIS) light spectrometer (J&M, Tidas, Essingen, Germany) equipped with a deuterium/halogen light source and a CCD detector.

##### Optical Characterization of Recombinant Spider Silk Films

Untreated and post-treated eADF4(C16) and eADF4(C16)-CBD film samples were attached to glass plates. The transmission was determined in the wavelength range from 173 nm to 997 nm with a resolution of 1 nm. At least 10 repeats per sample were performed at various positions on the respective film.

The propagation of the refractive index *n*(λ) was calculated by the fitting of the resulting transmittance curves to the curves simulated for physical models using the transfer-matrix method (TMM) [27]. For this process, the scattering, absorption and reflection were considered. The parameters of the models were consequently the thicknesses *d*, the concentrations of absorbers *C*_d_, the molar extinction coefficients *ε*(λ), the apparent concentrations of scatterers *C*_s_, and their size-dependent dispersion exponents *α* [15]. A micrometer gauge was used to determine *d*. The obtained thicknesses were refined within ±10%. Adjusted concentrations of the respective materials were referenced against their pure solvents for the determination of *ε*. The parameter *C*_d_ was estimated from the densities of the used materials. *n*(λ) for the TMM was modeled by the Sellmeier Equation (1):(1)$$n\left(\lambda \right)=\sqrt{A+B\lambda /\left(\lambda -C\right)}$$
with the Sellmeier coefficients *A*, *B* and *C* each independently refined. *C*_s_ and *α* were refined to approximate the scattering losses in thin films with the transmittance due to reflection and absorption *T*_a,r_ from TMM via (2).
(2)$$T_{\mathrm{tot}}=T_{a,r}-C_{s}/\lambda^{\alpha }$$

From each obtained refractive index curve, the Abbe numbers *ν*_D_ for the respective material could then be determined according to Equation (3).
(3)$$\nu_{D}=\left(n_{D}-1\right)/\left(n_{F}-n_{C}\right)$$
where *n*_F_, *n*_D_, *n*_C_ are the refractive indices at the wavelengths of 486 nm, 589 nm and 656 nm, respectively.

##### Attenuation Determination via Cut-Back Method

For the attenuation determination, the ends of the biopolymer optical fibers were connected to a bare fiber terminator and a multimode fiber connector of type SMA905 (Thorlabs GmbH, Bergkirchen, Germany). Afterwards, the fiber tips were aligned by using the SMA905 fiber stop (Thorlabs GmbH, Bergkirchen, Germany). After the optical fibers were fixed, their ends were cleaned using a lens cleaning tissue and a fiber optic splice and connector (Thorlabs GmbH, Bergkirchen, Germany). The radiation from the spectrometer was guided into the cellulose-based optical fibers via mono-optic fibers. For this purpose, SMA-to-SMA mating sleeves were used as the connection. The output power *I*_x_ of the fibers were measured from 173 nm to 997 nm with a 1 nm resolution. For each sample, the intensity *I*_2_ of three long fibers with the length *L*_2_ were measured. Afterwards the filaments were cut with a razor blade and the intensity *I*_1_ of the shorter fibers with a length of *L*_1_ were investigated. The attenuation loss *α* in dependance of the wavelength was determined by using Equation (4) [28,29].
(4)$$\alpha \left(\lambda \right)=10/\left(L_{2}-L_{1}\right)\cdot lg\left(I_{1}\left(\lambda \right)/I_{2}\left(\lambda \right)\right)$$

Both the intensities and the lengths of the long optical filaments were offset against the respective short filaments. The mean values and the associated standard deviations were accordingly calculated.

#### 2.2.8. Calculation of the Maximum Data Rate according to the Nyquist Rule

To calculate the maximum data rate, the numerical aperture (*NA*), the maximum time delay (Δ*t*_gmax_) and the bandwidth (*B*) are required. For the determination of *NA* the refractive index progressions of the core and cladding material are required. The data of the cellulose core were taken from our previous publication [15]. With the refractive indices determined in Section 2.2.7 it was possible to calculate the *NA*, Equation (5) [30]:(5)$$NA=\sqrt{n_{c}^{2}-n_{\mathrm{cl}}^{2}}$$
with *n*_c_ and *n*_cl_ as refractive index of the core and cladding material, respectively. The *NA* could then be used to determine the mode dispersion, which is Δ*t*_gmax_ between the longest and shortest rays through the core of the cellulose optical waveguide, Equation (6) [30]:(6)$$\Delta t_{\mathrm{gmax}}=L\cdot NA^{2}\cdot {\left(2c\cdot n_{c}\right)}^{-1}$$
with *L* as the length of the polymer optical fiber and *c* as the light velocity in vacuum. The relationship of mode dispersion and bandwidth could be approximated as follows, Equation (7) [31]:(7)$$B\approx 0.433/\Delta t_{\mathrm{gmax}}$$

According to the Nyquist theorem, the maximum data rate is $R_{\max}=2B\cdot {log}_{2}V$ with *V* as the number of discrete levels of the signal [30]. In our case it was *V* = 2, therefore *R*_max_ could be defined in bits per second (bps) as follows:(8)$$R_{\max}=1.722\cdot c\cdot n_{c}\cdot {\left(L\cdot NA^{2}\right)}^{-1}$$

#### 2.2.9. Infrared Spectroscopy

For the evaluation of the cladding layer, Fourier transform infrared (FTIR) spectra were generated from the optical fibers. For this purpose, a Frontier MIR spectrometer (L1280018) with an attenuated total reflection (ATR) diamond (PerkinElmer, Rodgau, Germany) was used. All spectra were obtained from eight scans in the wavenumber range of 400 to 4000 cm^−1^ with a resolution of 4 cm^−1^.

#### 2.2.10. Scanning Electron Microscopy (SEM) and Energy Dispersive X-ray Analysis (EDX)

SEM images were generated using a DSM 940A (Zeiss, Oberkochen, Germany). The manufactured optical fibers were mounted on aluminum holders with carbon adhesive discs. The samples were sputter-coated with gold/palladium. SEM imaging was performed at 15 keV electron energy, using the secondary electron detector. For the EDX measurements, a Quantax type 5 778 (Bruker Optics Inc., Billerica, Massachusetts, USA) was used at an acceleration voltage of 20 kV.

## 3. Results and Discussion

### 3.1. Characterization of Optical Properties of eADF4(C16) Films and Coatings

#### 3.1.1. Fabrication of eADF4(C16) Films and Determination of Optical Properties

Biocompatible optical materials are of high interest for medical applications. For this reason, cellulose has been chosen as the core and recombinant spider silk proteins as the cladding material. For light transmitting applications, it is necessary to know the optical properties such as the transmission and also the refractive index profile of both materials. Since the optical properties of RC were already determined in one of our publications, materials made of the silk protein eADF4(C16) were optically characterized accordingly [15]. For this purpose, it was fabricated appropriately into transparent films. HFIP was found to be a suitable solvent for this purpose. There, the proteins were dissolved overnight and then coagulated in cavities. In this process, HFIP favored the formation of an increased amount of alpha-helix secondary structures [32]. The resulting films exhibited high transmission. However, the coagulated films were water-soluble. Since this minimizes the application potential, these films had to be post-treated by using primary alcohols yielding to a general change in the secondary structure as already published [33,34]. The alpha-helix content was reduced and the beta-sheet content increased [35]. Therefore, a more dense material was produced also influencing the optical and mechanical properties of the silk films.

The intensity of the post-treatment, that is the concentration of alcohol in aqueous solution, can lead to turbidity of the material. If more ethanol is used, the material turns white and is brittle (80 wt% EtOH) as shown in Figure 1a. If the mass fraction of EtOH is reduced to about 70 wt%, a slight turbidity occurs. A further reduction leads to dissolution of the films due to the increased amount of water. It has been found that a mass fraction of about 75 wt% EtOH is an ideal condition for post-treatment of the silk proteins. Both the high transparency and the flexible properties are preserved.

The change in the secondary structure induced a change in the absorption behavior of the material. Transmission spectra were determined for films without and with post-treatment. Both had an absorption band at around 290 nm, Figure 1b. The non-post treated sample exhibited a strong absorption band. The transmission was reduced to 2.0%. The post-treatment decreased the losses, and 49.8% of the radiation was transmitted. Afterwards in the visible range of the spectrum, both films showed a high transparency. However, the post-treated sample achieved high transmission only at higher wavelengths. This might have been due to increased reflection at the interfaces of the films, which is also recognizable in Figure 1a. From the transmission profiles of the untreated and post-treated films, the refractive index progressions of the eADF4(C16) proteins could be determined via TMM, Figure 1c.

Here, the Abbe value of the untreated recombinant silk is in the range for previously measured spider silk [36]. In the spectral range of 200 nm to 1000 nm, the silk always had a lower refractive index than RC. Post-treatment in EtOH led to an increase in the refractive index as well as the Abbe value due to the denser structure induced by the increased beta sheet content [34]. In contrast to Aigner and Scheibel (2019), post-treatment was not performed with 70% EtOH for 2 h, instead for all samples at 75% for 1 min [33]. Nevertheless, the refractive index curve was still below the one of cellulose. From the refractive index curves obtained, the optical parameters such as the refractive indices *n*_F_, *n*_D_ and *n*_C_ of the wavelengths 486 nm, 589 nm and 656 nm respectively, as well as the Abbe number *ν*_D_ and the Sellmeier coefficients *A*, *B* and *C* could be determined, Table 1.

For lightwave-guiding, the core of an optical fiber should always has a higher refractive index than the cladding material. When light is introduced into the fiber, total reflection occurs at the interface between the core and cladding [28]. This means that the light is completely reflected and remains in the core to be transmitted. Therefore, the principle of total internal reflection was fulfilled for untreated as well as for post-treated eADF4(C16).

#### 3.1.2. Processing of Optical Fibers and Attenuation Determination

For the production of a core-cladding structure, the solution was applied dropwise to the fiber as illustrated in Figure 2a. The volatile solvent evaporated quickly, making it convenient to coat the cellulose-based filament. To prepare the RC-eADF4(C16) fibers for medical application, they were treated like the corresponding silk films with a 75 wt% EtOH solution and dried in a desiccator. Figure 2b shows an SEM image demonstrating that the fiber consisted of a circular core with a diameter of approximately 330 µm surrounded by a cladding layer varying between about 33 µm to 71 µm. The varying thickness of the silk coating is due to the coating process. However, clear delamination between the core and the cladding was observed. This prevented a permanent contact between the core and the cladding over the entire length of the fiber, which favors the formation of air voids. As a result, total reflection took place between the core and the cladding, as well as the core and the air. The post-treatment in the 75 wt% EtOH solution presumably led to a swelling of the cellulose filament due to the high hygroscopicity of cellulose [37]. The subsequent drying of these fibers caused the core to shrink and delamination occurred. The intermolecular interactions between eADF4(C16) and cellulose did not seem to be sufficient to preserve a permanent contact between the core and the cladding. As a result, the introduced radiation was decoupled from the core, and this part of the light was therefore no longer available for the selected use and led to an increase in attenuation.

To determine the attenuation, the fibers were examined using the UV/VIS spectrometer, and the intensities of the filaments were determined via the cut-back method, Figure 2c. The post-treated fibers, consisting of a RC core and an eADF4(C16) cladding, exhibited generally high attenuation. The light could be conducted above a wavelength of 520 nm. For lower wavelengths, the intensity was attenuated below the detectable limit. The attenuation then decreased and reached its minimum of 5.76 dB cm^−1^ at 878 nm. Subsequentlys, a slight band at 925 nm and a rise for another band at about 1000 nm could be detected. Comparing the attenuation of the RC core with the attenuation of the RC-eADF4(C16) core-cladding structure, this attenuation increase could be seen in the entire progression of the visible spectrum [15]. Since delamination affects the light waveguiding negatively, the use of such fibers is not advantageous.

### 3.2. Processing of eADF4(C16)-CBD for Improved Optical Fiber Properties

#### 3.2.1. Optimization of the Optical Fibers via Cellulose Binding Domain

To overcome the challenge of delamination, the recombinant silk proteins were genetically modified with a cellulose binding domain (CBD). In addition to the existing intermolecular interactions between the silk and the cellulose, the binding domain can actively bind the cellulose core and enhance the interactions. This should counteract the delamination and enable the use of the optical fibers.

However, since dissolution of the modified protein in organic solvents resulted in denaturation of the binding domain, rendering it non-functional, the modified recombinant silk had to be processed out of an aqueous medium. One challenge was that it was impossible to produce a suitable silk film from the aqueous medium thick enough for optical analysis. Multilayer films resulted in a white haze of the entire film. For that reason, no optical data such as the transmission or the refractive index progression, could be determined from these films. 

Therefore, the modified silk proteins were processed via HFIP for optical analysis as an approximation. Despite the inactive cellulose binding domain, the optical data of the untreated films, as well as the films post-treated with EtOH should represent the optical limiting properties. We assume that the optical data of the active eADF4(C16)-CBD proteins should be within the resulting range. This allowed us to reproduce the potential of eADF4(C16)-CBD proteins as a cladding material.

For the determination of the optical properties, eADF4(C16)-CBD was thus dissolved in HFIP and coagulated in cavities to form films. Since these films are subsequently water-soluble, they were also post-treated accordingly. 

#### 3.2.2. Determination of the Optical Properties of eADF4(C16)-CBD Films

The transmission spectra of eADF4(C16)-CBD films were determined from the obtained transparent films without and with post-treatment. As can be seen in Figure 3a, both materials have an absorption band at 290 nm such as the unmodified eADF4(C16). The untreated sample had a very narrow band, which caused the transmission to increase rapidly, and attained a value of 90% at 455 nm. Subsequently, the silk reached an average transmission of 92.3%. The post-treated sample had a broader absorption band. A transmission of 90% was reached above a wavelength of 798 nm. Subsequently, the transmission increased and attained a maximum value of 92.1% at a wavelength of 951 nm. A difference to the unmodified eADF4(C16) could be seen. Due to the modification with the CBD unit, the absorption band did not decrease after post-treatment. The following slow increase in transmission was similar to that of the unmodified variant. The refractive index progressions were then determined from the transmission profile via TMM, Figure 3b.

The untreated modified silk film exhibited a lower refractive index than the regenerated cellulose over the entire spectral range. The post-treatment and the change in the secondary structure also resulted in an increase in the refractive index profile. This could be caused by an increased beta sheet content. As a result, the structure adopted a more dense packaging, which led to an increase in the refractive index, similar to the unmodified variant. This resulted in the post-treated modified silk exhibiting a refractive index higher than that of the RC until 304 nm. Subsequently, the refractive index decreased further and was then below the course of the refractive index of RC. The Abbe value had not changed significantly in this case.

Since the cellulose fibers only conduct from a wavelength of about 500 nm, this result does not conflict with the principle of total internal reflection. Both the untreated and post-treated eADF4(C16)-CBD proteins could serve as cladding material for cellulose-based optical fibers. The most significant values, such as the refractive indices *n*_F_, *n*_D_, *n*_C_, the Abbe values *ν*_D_ and the Sellmeier coefficients *A*, *B* and *C,* were evaluated, Table 2.

#### 3.2.3. Fabrication of Core-Cladding Optical Fibers and Attenuation Determination

To coat the cellulose core with an eADF4(C16)-CBD layer, RC fibers were immersed in an aqueous protein solution, dried and post-treated in ethanol vapor. The fibers were characterized using SEM, EDX and FTIR to confirm the successful coating process. The SEM images (Figure 4) show a circular core with a diameter of about 330 µm surrounded by a fine cladding layer with a thickness of about 3.24 ± 0.22 µm. Delamination could not be observed using this aqueous coating method.

The elemental composition of the core material as well as the cladding layer was determined using EDX, see Table 3. Traces of sodium (Na), palladium (Pd) and gold (Au) could be detected in the core, as well as on the cladding layer. Both gold and palladium are artefacts from sputtering. Sodium could be a contaminant from water. The core consisted mainly of carbon (C) and oxygen (O). In the cladding, there was additionally nitrogen with an amount of 17.4 at.%.

FTIR analysis was performed on the pure core filament and the core-cladding-structured optical fiber surface, Figure 5. The obtained bands were shifted vertically from each other for better visual comparison.

The spectrum of the core filament had a broad band in the range from 3000 cm^−1^ to 3650 cm^−1^. This could be attributed to the valence vibrations of the hydroxyl groups within the cellulose chains, as well as to the vibrations of absorbed water, including signals from inter- and intramolecular interactions between the molecular chains [38]. The band around 2890 cm^−1^ could be assigned to the valence vibrations of alkyl groups. At 1639 cm^−1^ there was another band of absorbed water. The band in the region around 1429 cm^−1^ could be assigned to symmetrical deformation vibrations of CH_2_ groups. In the range between 1300 cm^−1^ and 1000 cm^−1^ the bands of C-O valence vibrations of the alcohols are located within the molecular chains. The stretching vibrations of the glycosidic bonds were located around 896 cm^−1^.

The FTIR spectrum of the cladding material showed Amide A (3250–3330 cm^−1^), Amide I (1600–1700 cm^−1^) and Amide II (1480–1575 cm^−1^) bands, typical for proteins [39]. The Amide I band primarily represents the C=O stretching vibrations of the amide backbone and the Amide II band displays the N-H bending in combination with C-N as-symmetric stretching vibrations [34]. Additional typical cellulose bands in the range around 850–1180 cm could also be seen.

The manufactured optical fibers were then examined using a UV/VIS spectrometer. The attenuation was calculated using Equation (4). The RC-eADF4(C16)-CBD fibers could guide light of different wavelengths, Figure 6a. In (i) and (ii) it can be seen that the scattering losses occurred exclusively in the coupling region area up to a length of about 7 cm. After that, no further scattering losses were visible to the human eye. According to Equation (4), the attenuation progression could be determined as a function of the wavelength, Figure 6b.

The RC core fiber conducted light at a wavelength of 500 nm. Subsequently, the attenuation decreased and reached a preliminary minimum at 734 nm with an attenuation of 0.72 dB cm^−1^. Afterwards, a weak band could be seen at 760 nm. Then, the attenuation decreased again and reached its absolute minimum of 0.60 dB cm^−1^ at a wavelength of 867 nm in the visible region of the spectrum. The attenuation increased again afterwards, with a medium band at 925 nm and a large band at 990 nm. Over the entire progression, the RC had a narrow standard deviation. The free-hanging drying process of the bioPOFs reduced the amount of extrinsic losses. This improved the uniformity of individual fibers, which leads to a reduction of the standard deviation of the basic attenuation of the cellulose compared to our previous work [15]. The cut-back method used here to determine the attenuation propagation yields to the same results as the attenuation values determined by the substitution method in our earlier publication.

Therefore, the RC fiber exhibited the typical attenuation spectrum of cellulose in the visible region of the spectrum [15]. The weak band around 760 nm is caused by the combination of the fifth and fourth harmonic absorption of the CH and OH groups of the cellulose. The shoulder at about 925 nm and the large band around 990 nm result from the 4th CH and 3rd OH harmonic vibration, respectively.

Upon coating the RC fibers with the spider silk protein containing the CBD, the core-cladding structure conducted the visible light only from a wavelength of 572 nm. Below that, the intensity was attenuated significantly beyond the detectable limit. The noise has been removed for more clarity in Figure 6b. The fibers had a similar attenuation progression as RC fibers. The attenuation was higher in the entire spectrum than for pure cellulose. There were also three bands visible: a weak band at 754 nm, a medium band at 925 nm and a large band at 990 nm. The absolute minimum was located at 878 nm with an attenuation of 1.91 dB cm^−1^. The standard deviation was higher compared to the pure RC fiber.

Considering the attenuation minimum, the attenuation was increased from 0.60 dB cm^−1^ to 1.91 dB cm^−1^. Therefore, compared to the unmodified eADF4(C16), the attenuation increase was reduced. The eADF4(C16)-CBD coating thus enables effective light conduction. The attenuation increase, as well as the loss of usable radiation from 500–572 nm was probably caused by the formation of further defects. Since the cladding process took place in an aqueous medium, the core fiber swelled and shrinked accordingly. A variety of defects, such as deformations and possibly cracks, could have occured. Scattering centers may have been formed, which scatter and decouple the low wavelength light and increase the basic attenuation. An optimization of the cladding process would reduce the basic attenuation, which would approximate the attenuation of the RC fibers.

### 3.3. Calculation of the Maximum Data Rate according to the Nyquist Rule

On the basis of the obtained refractive indices of the RC and the untreated and post-treated spider silk-CBD films, the *NA* of the respective core-cladding materials for the attenuation minimum of 878 nm could be determined using Equation (5). Afterwards, Equation (6) could be used to calculate the Δ*t*_gmax_ as a function of fiber length. Therefore, both the calculated *NA* and the vacuum speed of light of 2.9979·10^8^ m s^−1^ were used. The bandwidth *B* could be subsequently determined as a function of fiber length using Equation (7). This resulted in the maximum data rate *R_max_* via Equation (8), Figure 7.

The theoretical maximum data rate decreased rapidly within one meter of fiber length. When using a fiber consisting of RC as the core material and untreated eADF4(C16)-CBD as the cladding material, the data rate was below 10 Gbps at a fiber length of 0.48 m. When the post-treated silk was used as cladding material, the data rate was below 10 Gbps above a fiber length of 1.29 m.

Post-treatment with EtOH led to an increase in the refractive index. This reduced the *NA* and thus increased the usable length. It can therefore be assumed that the biocompatible fibers that have been cladded through an aqueous medium will fall under the 10 Gbps within this range of 0.48–1.29 m.

## 4. Conclusions

For a medical application, optical fibers must fulfill several requirements. They must be biocompatible and sterilizable, as well as providing high transmission. Both cellulose and spider silk are biocompatible compounds. The two materials are already used in a variety of medical applications such as wound dressings, tissue engineering and controllable drug delivery systems [40,41]. The manufactured fibers would be sterilizable by using EtOH, and all used materials exhibit high transmission. Possible standard applications in medicine include data transmission, sensing, endoscopic image guiding or illumination [42]. In order to serve a wide range of applications, an increasing amount of data has to be transmitted in a shorter time period because of increased digitalization. Based on this, the dependence of the theoretical data rate on the length of a manufactured fiber was calculated to fulfil the performance of a modern 10 Gbps Ethernet cable. The manufactured RC-eADF4(C16)-CBD fiber with a length between 0.48–1.29 m might be able to fulfill this requirement. Compared to fossil-based polymer optical fibers, this is a significant limitation of the usable length for technical applications [42,43,44]. In medical applications, however, only a few centimeters are usually required. Here, at least ~0.5 m of the optical fiber could be surgically introduced into human or animal tissue. The biocompatible fiber is in contact with living tissue, and outside of the tissue it could then be bridged with another polymer optical fiber, whether biocompatible or non-biocompatible.

In addition to biological sensing, this approach also allows the use of the optical fibers for optogenetic stimulation and photodynamic therapy (PDT). A major advantage is that cellulose-based optical fibers have the lowest attenuation in the wavelength range where the biological tissue has the highest penetration depth for the same wavelengths [45]. As a result, the optical fibers do not have to be implanted directly in the target tissue, but only a few millimeters in front of it. This avoids damaging any additional surrounding tissue. In the field of optogenetics, optical fibers could be used for the characteristic BphP1-PpsR2 optogenetic system, which is sensitive to 740–780 nm [46]. Thus, several types of cellular processes can be activated. In PDT, the common photosensitizers have an absorption peak around 600–700 nm [47]. Longer wavelength light is also preferred, because it can penetrate deeper into tissues, allowing the treatment of a larger target tissue. The light can be delivered from a laser, LED or lamp via an optical fiber to the photosensitizer. A reaction occurs between the light, the photosensitizer and oxygen to act specifically against cancer.

The biocompatible optical fibers consisting of cellulose and recombinant spider silk thus fulfil many requirements in the medical field. After the respective use, both the cellulose and the silk can be biologically degraded [48,49]. If the biodegradation is to be avoided, it is always possible to coat these bioPOFs with additional materials that either slow down or prevent the biodegradation.

The successful first-time combination of both materials as optical fibers could therefore have high application potential in the future due to the many beneficial properties mentioned. In this context, it is important to further optimize the fabrication of the cellulose fibers in order to reduce extrinsic attenuation effects. A suitable non-polar solvent system should be used for the coating process to prevent the cellulose from forming additional defects.

## Figures and Tables

**Figure 1 biomimetics-08-00037-f001:**
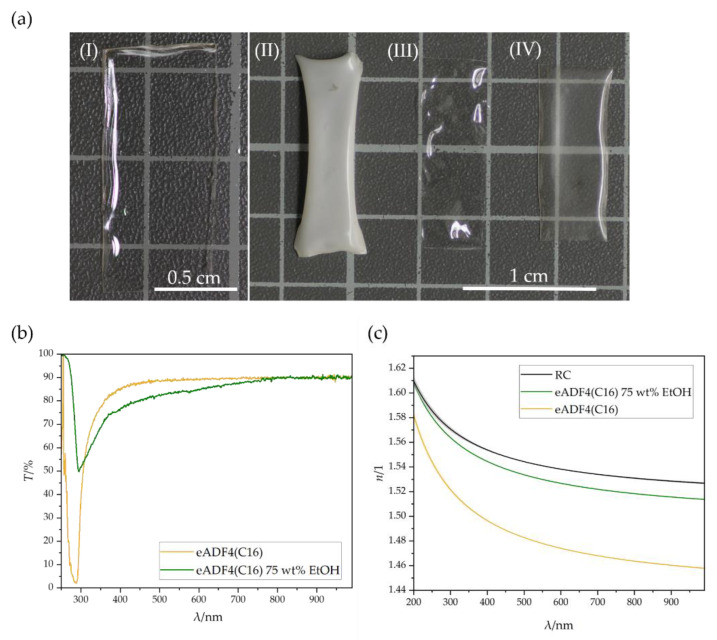
(**a**) Appearance of eADF4(C16) films (I) without post-treatment and post-treated with (II) 80 wt% EtOH, (III) 75 wt% EtOH and (IV) 70 wt% EtOH with a respective film thickness of 85 µm. (**b**) Transmission spectra of eADF4(C16) films, untreated and post-treated with 75 wt% EtOH. (**c**) Refractive index progression of RC, as well as untreated and post-treated eADF4(C16).

**Figure 2 biomimetics-08-00037-f002:**
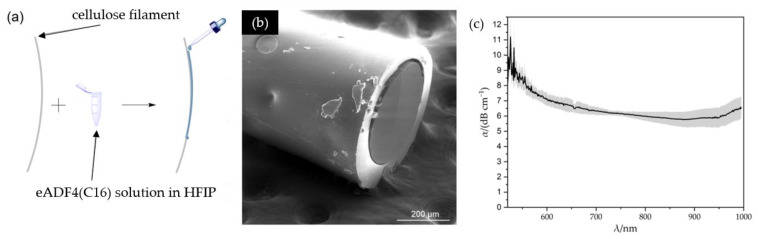
(**a**) Schematic process of manufacturing the core-cladding structure. (**b**) SEM image of the fiber tip. (**c**) Attenuation propagation of RC-eADF4(C16) optical fibers post-treated with 75 wt% EtOH.

**Figure 3 biomimetics-08-00037-f003:**
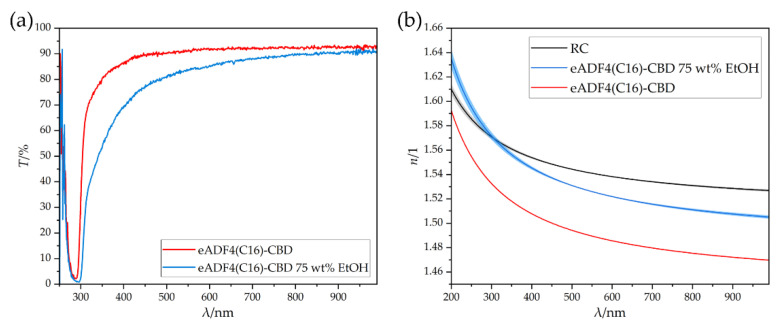
(**a**) Transmission spectrum of untreated eADF4(C16)-CBD and post-treated eADF4(C16)-CBD 75 wt% EtOH films with a respective film thickness of 85 µm. (**b**) Refractive index progression of regenerated cellulose, as well as untreated (eADF4(C16)-CBD) and post-treated (eADF4(C16)-CBD 75 wt% EtOH) spider silk protein films.

**Figure 4 biomimetics-08-00037-f004:**
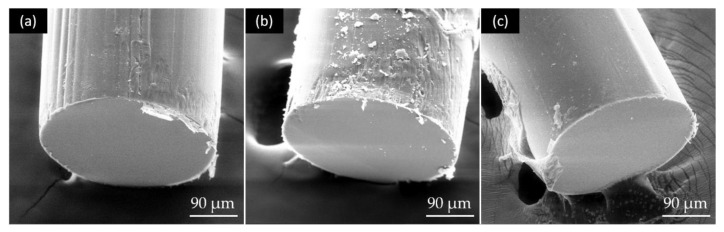
(**a**–**c**) SEM images of different fiber tips of the core-cladding structured optical fibers with RC as core material and eADF4(C16)-CBD as the cladding layer.

**Figure 5 biomimetics-08-00037-f005:**
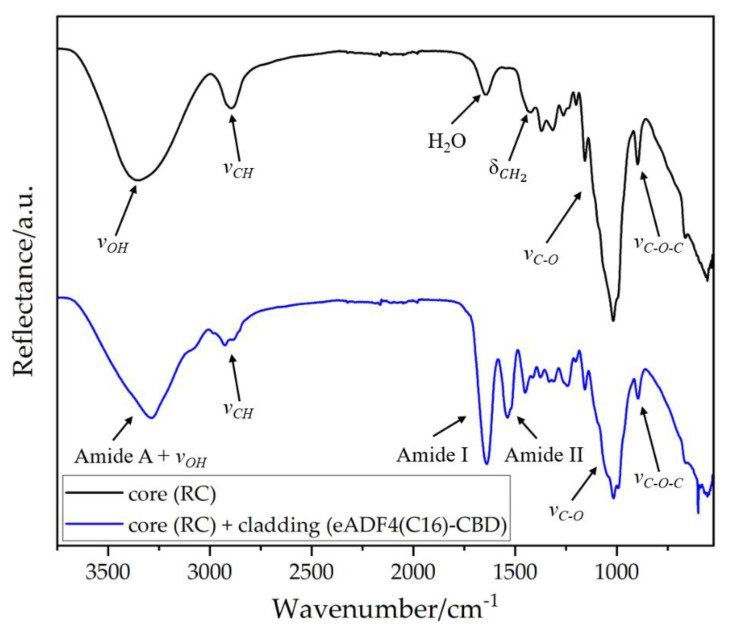
FTIR spectra of the RC core filament and the eADF4(C16)-CBD cladding layer.

**Figure 6 biomimetics-08-00037-f006:**
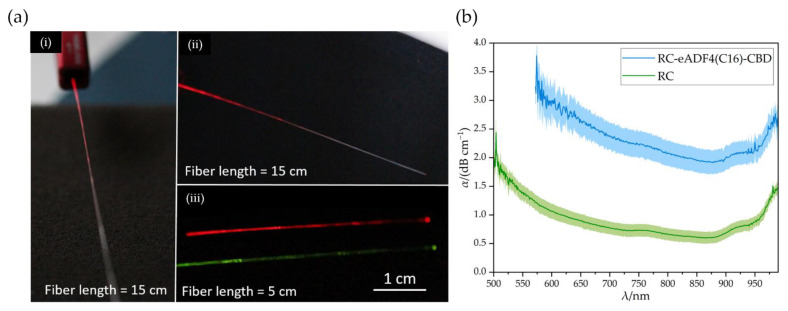
(**a**) BioPOFs consisting of RC as core and eADF4(C16)-CBD as cladding material. (i) and (ii) show a 15 cm long fiber segment at a wavelength of 670 nm. In (iii), 5 cm long fiber segments are shown at a wavelength of 538 nm (green) and 670 nm (red). (**b**) Attenuation progression of the pure RC core fiber and the RC-eADF4(C16)-CBD core-cladding structured optical fiber.

**Figure 7 biomimetics-08-00037-f007:**
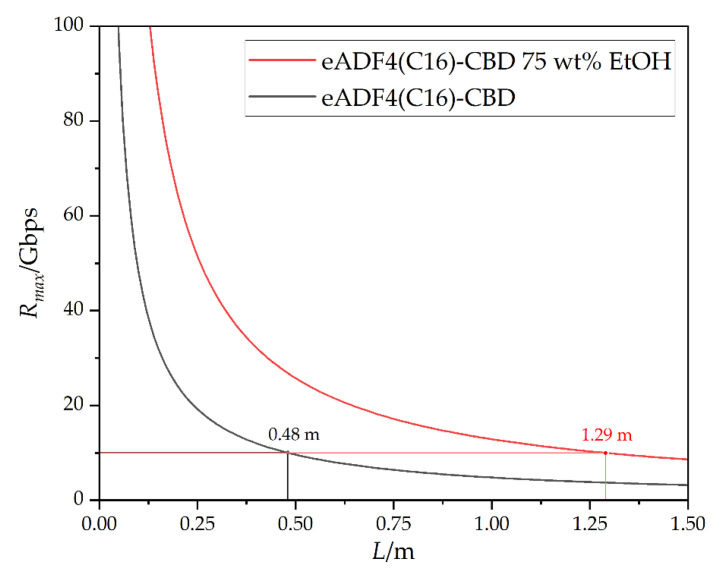
Maximum data rate of the biopolymer-optical fibers in dependence of the fiber length at a wavelength of 878 nm with a core diameter of 330 µm. For the mathematical calculation, the refractive indices of RC as the core material and of the untreated, as well as post-treated eADF4(C16)-CBD as the cladding are assumed, respectively.

**Table 1 biomimetics-08-00037-t001:** Refractive indices *n*_F_, *n*_D_ and *n*_C_ with standard deviation, the calculated Abbe values *ν*_D_ with the associated Gaussian error propagation, as well as the simulated Sellmeier coefficients *A*, *B* and *C* with standard deviation, obtained from the TMM simulation process from a triplicate determination. Data are displayed for RC, as well as untreated and post-treated eADF4(C16).

Sample	*n* _F_	*n* _D_	*n* _C_	*ν* _D_	*A*/1	*B*/1	*C*/nm
RC [15]	1.5454 ± 0.0002	1.5388 ± 0.0002	1.5357 ± 0.0004	55.65 ± 3.34	1.37 ± 0.05	0.91 ± 0.04	50.6 ± 0.7
eADF4(C16)	1.4843 ± 0.0002	1.4749 ± 0.0001	1.4705 ± 0.0002	34.25 ± 1.06	1.12 ± 0.05	0.94 ± 0.05	64.0 ± 2.5
eAD4(C16) 75 wt% EtOH	1.5349 ± 0.0001	1.52741 ± 0.00008	1.5239 ± 0.0001	47.76 ± 0.84	1.185 ± 0.009	1.050 ± 0.008	49.990 ± 0.006

**Table 2 biomimetics-08-00037-t002:** Refractive indices *n*_F_, *n*_D_ and *n*_C_ with standard deviation, the calculated Abbe values *ν*_D_ with the associated Gaussian error propagation, as well as the simulated Sellmeier coefficients *A*, *B* and *C* with standard deviation, obtained from the TMM simulation process from a triplicate determination.

Scheme 1.	*n* _F_	*n* _D_	*n* _C_	*ν* _D_	*A*/1	*B*/1	*C*/nm
eADF4(C16)-CBD	1.49574 ± 0.00006	1.48643 ± 0.00003	1.48210 ± 0.00006	35.66 ± 0.32	1.174 ± 0.005	0.922 ± 0.005	64.61 ± 0.21
eAD4(C16)-CBD 75 wt% EtOH	1.5326 ± 0.0004	1.5227 ± 0.0002	1.5182 ± 0.0005	36.18 ± 2.22	1.20 ± 0.04	1.00 ± 0.03	64.7 ± 1.5

**Table 3 biomimetics-08-00037-t003:** Elemental composition in at.% of the core and cladding material.

Elements	Core-Material	Cladding-Material
C	87.44	62.80
N	-	17.40
O	11.67	19.13
Na	0.12	0.18
Pd	0.16	0.09
Au	0.61	0.40

## Data Availability

The experimental data on the results reported in this manuscript are available upon reasonable request to the corresponding authors.

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
