# Peer review of "Biocompatible Optical Fibers Made of Regenerated Cellulose and Recombinant Cellulose-Binding Spider Silk"

_biomimetics, 2023, doi:10.3390/biomimetics8010037_

Round 1
Reviewer 1 Report
Reimer et al. present an interesting and well-written study detailing the production and characterization of core-sheath fibers produced with a regenerated cellulose core and recombinant spider silk cladding. These are demonstrated to have favorable behavior as optical fibers. As a whole, the results are well supported and robustly detailed. However, I do have a few minor concerns which I feel should be addressed prior to this being suitable for publication:
1) The new recombinant silk construct eADF4(C16)-CBD needs to be described in more detail, including the sequence of the fusion protein and the demonstration that it binds to cellulose and ability to process this protein. (Noted as “unpublished data” on line 142.) The subsequent discussion about the CBD having structuring that needs to be retained for its functionality is difficult to assess and evaluate on the basis of the information provided – knowing the size of this CBD would be helpful as part of this consideration and it would be beneficial to also provide information for the reader about the specific structural details of the CBD. Similarly, in section 3.2.1, a number of results statements are made about suitability (or lack of suitability) of various conditions for these fusion proteins without data to back them up.
2) Is there a specific optics-based rationale for the use of RC as the core and silk as the cladding? It would be helpful to provide and detail this, if so, or to note that there is not and detail the reason for this, if not.
3) The RC optical properties are being recapitulated from a previous study. How similar/dissimilar were the solvent conditions etc. under which these fibers were produced and the values determined?
4) In the para. spanning pp. 7-8, the wording and tense usage implies that the solvent-dependent structural inferences noted for the eADF4(C16) are in the present study; however, it appears that these are all in fact from prior studies. This needs to be clarified and if indeed from prior studies the degree of similarity (or lack thereof) in conditions needs to be explicitly spelled out.
5) In Figure 1(a), why is an untreated film not shown for direct comparison?
6) Is eADF4(C16) suitable for deposition in a similar manner to eADF4(C16)-CBD? It would be ideal to have a side-by-side comparison of cladding produced using both silks under the same conditions, instead of very disparate conditions.
7) In the FTIR data shown in Fig. 5, the spectra and features of the fiber with cladding are almost always referred to as coming from just the cladding. However, unless I am mistaken, these are actually for the RC core fiber with cladding. Since at least some FTIR bands of RC are still seen (i.e., the ATR is not showing *only* features of the cladding), this should be clarified to be cladding + core. It would also be beneficial to annotate the key bands directly on Fig. 5, instead of just in the text.
8) In the conclusion, it’s noted that EtOH could be used to sterilize the materials. Isn’t this contrary to the EtOH-induced changes in structuring and properties that are noted earlier in the manuscript?
9) There are a number of minor typographical/language/contextual issues that would benefit from being corrected:
Line 55: “For *the* optical fibers” – which fibers specifically are *the* fibers?
Line 64 and later: need to define bioPOF
Para. spanning pp. 2-3: it would be beneficial to introduce and summarize some (recombinant) spider silk materials and optical properties, rather than strictly those of silkworm silks
Line 130: what’s meant by the fiber being “wound up” at constant speed?
Line 184: should this be “the” Sellmeier equation rather than “a”?
Line 314: “obviously” seems like a subjective conclusion?
Line 346: better wording may be “the binding domain can actively”
Lines 358-359: what’s meant by “should represent the optical limiting properties”
Line 360: there appears to be a missing work between “allowed” and “to”
Line 443: the data shown in Figure 6 appear to be from more than strictly UV/Vis spectroscopy?
Line 514: likely better as “in medicine” than “in the medicine”
Line 515: reword “more and more data”
Line 520: missing word with “fossil-based”?
Author Response
Dear Ms Qin, dear reviewers,
Thank you for your comments. We have tried to implement all to the best of our knowledge according to the specifications. Below you will find the answers to your respective comments.
Answer to editor:
The sentences marked in the material and methods part have been reworded accordingly.
Reviewer 1:
1.
The new recombinant silk construct eADF4(C16)-CBD needs to be described in more detail, including the sequence of the fusion protein and the demonstration that it binds to cellulose and ability to process this protein. (Noted as “unpublished data” on line 142.) The subsequent discussion about the CBD having structuring that needs to be retained for its functionality is difficult to assess and evaluate on the basis of the information provided – knowing the size of this CBD would be helpful as part of this consideration and it would be beneficial to also provide information for the reader about the specific structural details of the CBD. Similarly, in section 3.2.1, a number of results statements are made about suitability (or lack of suitability) of various conditions for these fusion proteins without data to back them up.
- The “unpublished data” refers to a stand-alone publication on eADF4(C16)-CBD which is currently in the editing phase. In this publication, quartzcrystal microbalance measurements have been conduced to demonstrate the cellulose-binidng capabilities. Further details about the utilized CBD can be found in (Ong et al. 1993), which now has been indicated more clearly.
Is there a specific optics-based rationale for the use of RC as the core and silk as the cladding? It would be helpful to provide and detail this, if so, or to note that there is not and detail the reason for this, if not.
- Light waveguiding in an optical fiber takes place due to total internal reflection. To achieve this, the core must have a higher refractive index than the cladding. This is explained before and after Table 1 on page 7-8.
3.
The RC optical properties are being recapitulated from a previous study. How similar/dissimilar were the solvent conditions etc. under which these fibers were produced and the values determined?
- The optical data (refractive index profile, Abbe value) were determined from regenerated cellulose films. The dissolution process of the microcrystalline cellulose in the binary solvent system DMAc/LiCl is exactly the same as described in the publication (Reimer et al. 2021), as described in the Material and Methods part “2.2.1 Dissolution of MCC”.
“MCC was dissolved in DMAc/LiCl according to our previous publication [15].”
For the production of the fibers, the wetspinning process was slightly modified. This is also described in 2.2.2 „Wetspinning of regenerated cellulose fibers“.
„Wet spinning was performed according to our previous publication with exception of the flow rate of the pump and the drying [15]”.
The modification of the manufacturing process resulted in a reduction of the standard deviation. This difference in light conduction is now highlighted in the following text. In contrast to our previous publication, the cut-back method was used instead of the substitutions method to determine the attenuation. However, both methods provide the same mean values of the attenuation profile.
In the para. spanning pp. 7-8, the wording and tense usage implies that the solvent-dependent structural inferences noted for the eADF4(C16) are in the present study; however, it appears that these are all in fact from prior studies. This needs to be clarified and if indeed from prior studies the degree of similarity (or lack thereof) in conditions needs to be explicitly spelled out.
- The post-treatment of silk with alcohols leads to a general change in the secondary structure, which is known from the literature. In general, the alpha helix content is reduced and the beta sheet content is increased. This occurs both when silk is post-treated by steam or with a liquid. The reference to literature has been highlighted. The difference in the post-treatment are now mentioned.
5) In Figure 1(a), why is an untreated film not shown for direct comparison?
- The water solubility of the untreated silk films means that technical use is difficult. However, since the reader might be interested in the optical differences, which are recognizable to the naked eye, an image of the untreated eADF4(C16) film has been added to Figure 1.
6)
Is eADF4(C16) suitable for deposition in a similar manner to eADF4(C16)-CBD? It would be ideal to have a side-by-side comparison of cladding produced using both silks under the same conditions, instead of very disparate conditions.
- The general purpose is to produce an optical fiber consisting of cellulose as the core material and silk as the cladding layer. Since cellulose is a strongly hygroscopic material, a swelling process occurs on contact with water. Subsequent drying leads to shrinkage of the fiber. The swelling and shrinkage in turn leads to the formation of defects and reduces the quality of the optical fiber. Since eADF4(C16) can be applied to the cellulose filaments via the volatile solvent HFIP, this approach was used to avoid the formation of potential defects within the cellulose core. Nevertheless, post-treatment with 75 wt% EtOH subsequently resulted in delamination due to the swelling process. If, in addition, the coating process were to take place in an aqueous solution, the attenuation would be significantly increased, which in turn minimizes the application potential. Therefore, it is also pointed out at the end of the Conclusions that non-polar solvents should be used for the coating process.
7)
In the FTIR data shown in Fig. 5, the spectra and features of the fiber with cladding are almost always referred to as coming from just the cladding. However, unless I am mistaken, these are actually for the RC core fiber with cladding. Since at least some FTIR bands of RC are still seen (i.e., the ATR is not showing *only* features of the cladding), this should be clarified to be cladding + core. It would also be beneficial to annotate the key bands directly on Fig. 5, instead of just in the text.
- That’s right. The designation „cladding + core" was added. The main bands are also marked in the figure.
8)
In the conclusion, it’s noted that EtOH could be used to sterilize the materials. Isn’t this contrary to the EtOH-induced changes in structuring and properties that are noted earlier in the manuscript?
- For disinfection, 75 wt% EtOH could be used specifically. If not wanted, an additional layer of eADF4-CBD can be applied after the production of the optical fibers. Since the total reflection then only takes place between the cellulose core and the first silk layer, the second cladding could also be treated with 70 wt% EtOH if necessary.
9) There are a number of minor typographical/language/contextual issues that would benefit from being corrected:
Line 55: “For *the* optical fibers” – which fibers specifically are *the* fibers?
- “The” was removed.
Line 64 and later: need to define bioPOF
- bioPOF is now defined
Para. spanning pp. 2-3: it would be beneficial to introduce and summarize some (recombinant) spider silk materials and optical properties, rather than strictly those of silkworm silks
- Until now, Bombyx mori silk was mainly used for the production of optical materials.
Line 130: what’s meant by the fiber being “wound up” at constant speed?
- Constant speed was removed. The fiber was wound up should mean that we roll them up and collect them.
Line 184: should this be “the” Sellmeier equation rather than “a”?
- The sentence was corrected.
Line 314: “obviously” seems like a subjective conclusion?
- “obviously” was removed.
Line 346: better wording may be “the binding domain can actively”
- The sentence was corrected.
Lines 358-359: what’s meant by “should represent the optical limiting properties”
- These should be the limiting properties in which the optical properties of the modified silk should lie. As it is also explained in the following sentence in our work.
Line 360: there appears to be a missing work between “allowed” and “to”
- The sentence was corrected
Line 443: the data shown in Figure 6 appear to be from more than strictly UV/Vis spectroscopy?
- An additionally inserted sentence should now provide clarity.
Line 514: likely better as “in medicine” than “in the medicine”
- The sentence was corrected
Line 515: reword “more and more data”
- The sentence was reworded
Line 520: missing word with “fossil-based”?
- The sentence should fit as specified.

Reviewer 2 Report
The manuscript reports on the fabrication of organic fibers as optical waveguides for medical and biological applications.
The manuscript is well organized and written, placed in a proper context and the basic ideas clearly stablished. The process of fabrication of cellulose/silk fibers is explained in detail and the results could be reproduce by specialized laboratories. Although the attenuation of the fibers is still high, the development of new materials for fiber fabrication might disclose unexpected uses.
I have a few comments that the authors should consider:
1.- I miss a presentation of the work carried out in the introduction and a short discus of what is new in this work with respect to previously published papers.
2. Please make a comment about the life time of this kind of fibers.
3. The thickness of the films used to obtain figures 1 and 3 must be given somewhere (for example in the figure caption).
4. The units of parameters B and C must be given in tables 1 and 2.
5. It must be specified if the composition of core and cladding materials are given in weight or in mol.
Author Response
Dear Ms Qin, dear reviewers,
Thank you for your comments. We have tried to implement all to the best of our knowledge according to the specifications. Below you will find the answers to your respective comments.
Answer to editor:
The sentences marked in the material and methods part have been reworded accordingly.
Response to Reviewer 2:
1.
I miss a presentation of the work carried out in the introduction and a short discus of what is new in this work with respect to previously published papers.
- The combination of both materials (cellulose, recombinant spider silk) is sparsely described in the literature as mentioned in the indroduction. The innovation in this work is the first successful combination of these compounds for the fabrication of optical fibers. This is now highlighted at the end of the introduction.
Please make a comment about the life time of this kind of fibers.
- The biodegradability is now mentioned in the conclusion section.
The thickness of the films used to obtain figures 1 and 3 must be given somewhere (for example in the figure caption).
- The film thickness was added.
- The units of parameters B and C must be given in tables 1 and 2.
- The parameters A and B are unitless. C hast the unit nm. The units were added in the tables.
- It must be specified if the composition of core and cladding materials are given in weight or in mol.
- The parameters for the production were given in detail in the material and methods section. In addition the thickness of the cladding layer was determined after the coating the process.

Round 2
Reviewer 1 Report
I still would have preferred a direct citation to the upcoming work that shows the CBD functionality and/or to see a draft of the manuscript in question as this is critical to the work being described here; however, I am willing to give the authors the benefit of the doubt.